# Soloxolone Methyl Reduces the Stimulatory Effect of Leptin on the Aggressive Phenotype of Murine Neuro2a Neuroblastoma Cells via the MAPK/ERK1/2 Pathway

**DOI:** 10.3390/ph16101369

**Published:** 2023-09-27

**Authors:** Kirill V. Odarenko, Oksana V. Salomatina, Ivan V. Chernikov, Nariman F. Salakhutdinov, Marina A. Zenkova, Andrey V. Markov

**Affiliations:** 1Institute of Chemical Biology and Fundamental Medicine, Siberian Branch of the Russian Academy of Sciences, 630090 Novosibirsk, Russia; k.odarenko@yandex.ru (K.V.O.); ana@nioch.nsc.ru (O.V.S.); chernikovivanv@gmail.com (I.V.C.); marzen@niboch.nsc.ru (M.A.Z.); 2N.N. Vorozhtsov Novosibirsk Institute of Organic Chemistry, Siberian Branch of the Russian Academy of Sciences, 630090 Novosibirsk, Russia; anvar@nioch.nsc.ru

**Keywords:** leptin, neuroblastoma, adipokines, aggressiveness, triterpenoids, EMT, molecular docking

## Abstract

Despite the proven tumorigenic effect of leptin on epithelial-derived cancers, its impact on the aggressiveness of neural crest-derived cancers, notably neuroblastoma, remains largely unexplored. In our study, for the first time, transcriptome analysis of neuroblastoma tissue demonstrated that the level of leptin is elevated in neuroblastoma patients along with the severity of the disease and is inversely correlated with patient survival. The treatment of murine Neuro2a neuroblastoma cells with leptin significantly stimulated their proliferation and motility and reduced cell adhesion, thus rendering the phenotype of neuroblastoma cells more aggressive. Given the proven efficacy of cyanoenone-bearing semisynthetic triterpenoids in inhibiting the growth of neuroblastoma and preventing obesity in vivo, the effect of soloxolone methyl (SM) on leptin-stimulated Neuro2a cells was further investigated. We found that SM effectively abolished leptin-induced proliferation of Neuro2a cells by inducing G1/S cell cycle arrest and restored their adhesiveness to extracellular matrix (ECM) proteins to near control levels through the upregulation of vimentin, zonula occludens protein 1 (ZO-1), cell adhesion molecule L1 (*L1cam)*, and neural cell adhesion molecule 1 (*Ncam1)*. Moreover, SM significantly suppressed the leptin-associated phosphorylation of extracellular signal-regulated kinase 1/2 (ERK1/2) and ribosomal protein S6 kinase A1 (p90RSK), which are key kinases that ensure the survival and proliferation of cancer cells. Further molecular modeling studies demonstrated that the inhibitory effect of SM on the mitogen-activated protein kinase (MAPK)/ERK1/2 signaling pathway can be mediated by its direct interaction with ERK2 and its upstream regulators, son of sevenless homolog 1 (SOS) and mitogen-activated protein kinase kinase 1 (MEK1). Taken together, our findings in murine Neuro2a cells provide novel evidence of the stimulatory effect of leptin on the aggressiveness of neuroblastoma, which requires further detailed studies in human neuroblastoma cells and relevant animal models. The obtained results indicate that SM can be considered a promising drug candidate capable of reducing the impact of adipokines on tumor progression.

## 1. Introduction

Neural crest (NC) cells possess specific molecular and epigenetic characteristics that enable them to produce diverse cell types. Following embryogenesis, these mechanisms are deactivated, but they can be reactivated in NC descendants that accompany the malignant transformation of neuroblastoma, melanoma, ganglioneuroma, medullary thyroid carcinoma, and pheochromocytoma. Tumor cells originating from NC cells regain many of their characteristics, including the transition from an epithelial to a mesenchymal phenotype, invasiveness, motility, and pluripotency, which are strongly linked to tumor aggressiveness [1].

Neuroblastoma is a pediatric malignancy of the sympathetic nervous system that is diagnosed in children with an average age of 17.3 months and accounts for nearly 15% of cancer-related infant deaths [2]. According to Zhou et al., the majority of low- and intermediate-risk neuroblastoma patients can be cured, while high-risk patients have poor outcomes, with 5-year survival rates below 50% [3]. Despite the significant advances in radiation therapy and the development of novel therapeutics (e.g., anti-CD2 monoclonal antibodies), approximately half of high-risk neuroblastoma patients do not respond to first-line treatment or experience relapse within two years after treatment [3]. These data indicate the need for further exploration of crucial factors underlying the aggressiveness of neuroblastoma and the development of novel antitumor agents targeting these factors.

Obesity is known to increase the incidence and severity of a number of cancers [4]. Despite the proven stimulatory effect of obesity on the aggressiveness of melanoma [5], its influence on other NC-derived tumors is poorly investigated. Given the frequent association of rapid-onset obesity with hypothalamic dysfunction, hypoventilation, and autonomic dysregulation (ROHAD) syndrome with neural tumors [6], as well as the recently reported stimulatory effect of neuroblastoma treatment on the overgrowth of visceral and subcutaneous adipose tissue in patients [7], there is a regulatory relationship between obesity and neuroblastoma growth that requires further detailed studies.

Adipose tissue produces a number of adipokines displaying modulatory effects on surrounding cells, the most extensively studied of which is leptin, which stimulates tumor growth, metastasis, inflammation, and angiogenesis [8,9]. To our knowledge, only Tümmler et al. have addressed the effect of adipokines on neuroblastoma. It was found that chemerin promotes the proliferation, clonogenicity, and xenograft growth of neuroblastoma cells through the MAPK/ERK and Akt pathways [10]. Moreover, the expression of the chemerin receptors CMKLR1 and GRP1 was found to have an inverse correlation with the survival of neuroblastoma patients [10]. The impact of other adipokines on the progression of neuroblastoma has not been specifically addressed. The available data only demonstrated the effect of leptin on cell proliferation and its protective potency in neuroblastoma cells against a range of cytotoxic stimuli [11,12].

Natural compounds make vital contributions to drug development due to their structural diversity and complexity unattainable by synthetic chemistry methods. Several preclinical studies have identified plant-derived compounds that inhibit cancer growth by targeting leptin signaling, including honokiol [13], silibinin, and curcumin [14]. Interestingly, triterpenoids with significant antitumor potential have not yet been evaluated as inhibitors of the pro-tumor activity of leptin, despite the proven ability of ursolic acid and escin to effectively suppress leptin-mediated processes in various non-tumor models, such as atherosclerosis [15] and high-fat diet-induced obesity [16], respectively.

Previously, our research group developed soloxolone methyl (SM), a cyanoenone-bearing semisynthetic derivative of glycyrrhetinic acid (Figure 1). This compound was shown to significantly inhibit the viability of NC-derived tumor cells [17] as well as the epithelial-mesenchymal transition (EMT) of melanoma B16 cells and their metastasis in vivo [18]. Considering the revealed antitumor potency of SM and the ability of 2-cyano-3, 12-dioxooleana-1, 9-dien-28-imidazolide (CDDO-Im) (a structural analog of SM; a semisynthetic derivative of oleanolic acid) to prevent body weight gain in high-fat diet-induced obese mice [19], the effect of SM and its analogs on adipokine-induced tumor cells is worthy of detailed investigation.

In light of these data, our study aimed to address two specific questions: first, whether leptin is capable of increasing the aggressiveness of neuroblastoma, and second, whether cyanoenone-bearing triterpenoids can suppress the tumor-promoting effect of leptin on neuroblastoma cells. The influence of leptin on the proliferation, motility, morphology, adhesiveness, and neuritogenic capacity of mouse neuroblastoma Neuro2a cells was explored. Next, the inhibitory effect of SM and its derivatives on key pro-tumor characteristics of leptin-stimulated Neuro2a cells was studied. Although these experiments were performed only in mouse cells, the results obtained provide valuable insights into the relationship between leptin and neuroblastoma progression that need to be further explored in human neuroblastoma cells. In addition, our findings also support the rationale for further investigation of SM and its analogs as potential drug candidates capable of affecting obesity-induced tumor growth.

## 2. Results

### 2.1. Leptin Promotes Aggressiveness in Neuroblastoma Cells

Given the previously reported stimulating effects of leptin on various malignancies, including NC-derived melanoma [20] and pheochromocytoma [21], first, we questioned whether leptin has pro-tumor functions in neuroblastoma. To address this issue, cDNA microarray data from 498 neuroblastoma tumor samples (GSE49710) were reanalyzed. It was found that the median expression of leptin in tumor nodes increased with the progression of neuroblastoma (Figure 2A). Interestingly, patients with International Neuroblastoma Staging System (INSS) stage 4S, which is characterized by a favorable outcome [22], demonstrated lower expression of leptin compared to INSS stage 4 high-risk neuroblastoma patients (Figure 2A). Performed Kaplan–Meier survival analysis demonstrated a significant correlation between high leptin mRNA level and worse overall survival in neuroblastoma patients (Figure 2B), which could indicate the ability of leptin to enhance the malignant characteristics of neuroblastoma.

To experimentally verify the results of transcriptome analysis, we evaluated the impact of leptin on several characteristics associated with the aggressiveness of neuroblastoma cells, including proliferation rate, motility, changes in cellular morphology, adhesiveness, and modulation of EMT markers. Leptin concentrations ranging from 200 to 800 ng/mL, corresponding to the serum level of leptin in mouse models of obesity [23,24,25], were tested in this study. As depicted in Figure 2C, the treatment of murine Neuro2a cells with leptin dose-independently increased cell growth by approximately 15% compared to the untreated control, which is in agreement with previously published results [26]. A further wound healing assay demonstrated that leptin significantly stimulated the motility of Neuro2a cells. The incubation of the cells with leptin at 400 ng/mL increased the rate of cell migration towards wounds by 33.6% compared to the control; further two-fold decrease in leptin concentration did not enhance this effect (Figure 2D). Evaluation of cell morphology showed no influence of leptin on either cell area (Figure 2E and Appendix A) or neurite outgrowth (Figure 2F and Appendix A), indicating the lack of differentiation-inducing activity of leptin in Neuro2a cells.

Considering that low adhesion strength is a key characteristic of metastatic cells [27], we further assessed the effect of leptin on cell adhesiveness. It was found that pre-treating Neuro2a cells with leptin for 24 h resulted in a 2.7-fold reduction in their adhesion to Matrigel-coated wells (Figure 2G). To explore this effect more precisely, the influence of leptin on the expression of key adhesion-associated proteins that play a significant role in EMT, namely ZO-1 (an epithelial marker), vimentin, and fibronectin (mesenchymal markers), was evaluated. Performed flow cytometry analysis revealed that the treatment of Neuro2a cells with leptin led to the upregulation of ZO-1 and marked suppression of vimentin expression, while fibronectin was unaffected by leptin induction (Figure 2H). Given that vimentin is a crucial regulator of the β1 integrin adhesive machinery [28], the observed downregulation of vimentin induced by leptin could explain its inhibitory effect on the adhesiveness of neuroblastoma cells. Notably, despite the proven ability of leptin to promote the acquisition of epithelial-derived tumor cells by mesenchymal traits (induction of EMT) [29], in NC-derived neuroblastoma cells, this adipokine induced a shift of EMT markers towards an epithelial phenotype, as shown in Figure 2H. Despite this finding, the pro-mitogenic (Figure 2C), pro-migratory (Figure 2D), and adhesion-suppressive (Figure 2G) effects of leptin on Neuro2a cells clearly indicate its stimulating influence on neuroblastoma aggressiveness.

### 2.2. Cyanoenone-Bearing Triterpenoids Effectively Inhibit the Proliferative Response of Neuroblastoma Cells to Leptin

In the next step of the study, we explored the effect of SM and its derivatives on the growth-stimulating property of leptin in neuroblastoma cells. To additionally evaluate the impact of the substituent at C-30 of SM on this activity, SM derivatives, S and SAO, containing a hydroxyl group and a bulky pivalic amidoxime group at C-30, respectively (Figure 1), were included in the experiment. First, non-toxic concentrations of the studied triterpenoids were evaluated in Neuro2a cells. The performed MTT assay demonstrated that the compounds did not cause any reduction in cell viability at 0.1 µM, whereas further increase in their concentrations led to a moderate suppression of cell proliferation (Figure 3A). Based on the obtained results, a triterpenoid concentration of 0.1 µM was selected for further analysis.

It was found that SM effectively suppressed the mitogenic effect of leptin in Neuro2a cells, reducing the proliferation of leptin-treated cells to almost the same level as the control (Figure 3B). Interestingly, substituents at C-30 of the evaluated triterpenoids had no obvious effect on this activity; both S and SAO showed potency similar to that of SM (Figure 3B). Notably, the observed effect of the tested compounds was cell line independent. The triterpenoids at a non-toxic concentration of 0.1 µM significantly inhibited the mitogenic effect of leptin in B16 melanoma cells also, showing an efficacy similar to that observed in neuroblastoma cells (Figure 3C,D). Considering the comparable bioactivities of the explored compounds (Figure 3B,D), SM, previously reported as an effective blocker of melanoma B16 aggressiveness [18], was selected for further studies.

Cell cycle analysis revealed that leptin caused a slight accumulation of Neuro2a cells in the S phase compared to the control (Figure 3E). The treatment of leptin-stimulated cells with SM resulted in the blockage of the G1/S cell cycle transition, since the enrichment of the G1-phase fraction and the reduction of cell populations in the S and G2 phases by 31%, 42%, and 61%, respectively, were identified in SM-treated cells compared to the leptin-induced group (Figure 3E). We hypothesize that the observed SM-induced cell cycle arrest (Figure 3E) could underlie the suppressive effect of SM on the mitogenic activity of leptin described above (Figure 3B).

### 2.3. SM Improves the Adhesiveness of Neuroblastoma Cells Stimulated with Leptin

Next, we asked whether SM could modulate leptin-induced changes in the metastasis-associated characteristics of neuroblastoma cells. The performed wound healing assay demonstrated that SM did not affect the leptin-driven increase in Neuro2a motility (Figure 4A). However, it completely abrogated the suppressive effect of leptin on cell adhesiveness to the Matrigel-coated surface (Figure 4B). To explore the latter more precisely, the adhesion capacity of Neuro2a cells to other surfaces, such as tissue culture-treated polystyrene and collagen-coated wells, was analyzed. As illustrated in Figure 3B, leptin significantly decreased the adhesiveness of Neuro2a cells in all groups, regardless of the tested surfaces, and SM markedly suppressed this effect. We hypothesized that the observed pro-adhesive properties of SM may be the result of its modulatory effect on the expression profile of cell adhesion molecules (CAMs). Indeed, RT-PCR analysis demonstrated that SM upregulated the expression of *L1cam* and *Ncam1* in leptin-induced Neuro2a cells, although the effect was relatively weak (Figure 4C). Surprisingly, the expression of *Itga1* was reduced in leptin-stimulated cells in response to SM treatment (Figure 4C). Notably, the leptin-induced loss of cellular adhesiveness was not associated with a decrease in the mRNA level of CAMs-related genes (Figure 4C). This discrepancy can be explained by examining gene expression at the late time point (48 h), when the effect of leptin on CAMs may be more pronounced at the protein level. Indeed, further flow cytometry analysis demonstrated marked reduction in vimentin expression following the treatment of Neuro2a cells with leptin (Figure 4D). SM was found to abrogate this effect and obviously enhance the expression of ZO-1 in leptin-stimulated neuroblastoma cells (Figure 4D).

### 2.4. SM Inhibits the Induction of the ERK1/2 Pathway by Leptin

The ERK1/2 mitogen-activated protein kinase is a well-known downstream effector of leptin, which regulates cell proliferation and migration [30]. Given the ability of pentacyclic triterpenoids to inhibit ERK1/2 activation in various tumor cells [31], we assessed whether SM affects the phosphorylation level of ERK1/2 in Neuro2a cells (Appendix A). As depicted in Figure 5A, a 30 min leptin induction caused an increase in the phosphorylation of ERK1/2 and its downstream effector p90 ribosomal S6 kinase (p90RSK) by 51.1% and 49.2%, respectively. SM completely abolished the stimulating effect of leptin on the ERK1/2/p90RSK signaling axis (Figure 5A).

To shed light on the probable molecular mechanism of this activity, we explored the binding affinity of SM to ERK1/2 and its key upstream regulators through molecular docking. The obtained results indicated that SM could interact with the nucleotide exchange factor SOS, as well as MEK1 and ERK2 kinases, with binding energies comparable to those of their known inhibitors (Figure 5C–E). As depicted in Figure 5E:The SM binding pocket in SOS was similar to that of the known SOS inhibitor CHEMBL4166974 and contained Pi-alkyl and Pi-sigma interactions with Tyr884, Lys898, and Phe890, which were involved in a hydrophobic pocket at the contact surface of SOS and H-Ras [32].SM can bind to the MEK1-KSR2 heterodimer, forming a hydrogen bond with Ser222, which is a key phosphorylation site of MEK1 [33]. Additionally, it forms two hydrogen bonds with Asn826 and Ala879 in KSR2, the first of which is crucial for the correct interaction of KSR2 with MEK1 [34].In the case of ERK2, SM formed two hydrogen bonds with Lys54, which is involved in the catalytic K/D/D motif [35], and Met108, a well-known inhibitor-binding residue in ERK2 [36].

Thus, the obtained results demonstrate that SM effectively inhibits the leptin-susceptible MAPK/ERK1/2 pathway, theoretically through its direct interactions with SOS, MEK1, and ERK2. Notably, these findings only provide predictive insights into the molecular mechanism of action of SM, which requires further experimental verification.

## 3. Discussion

Cancer is a complex multifactorial disease, the aggressiveness of which can be significantly enhanced by various factors, including adipokines secreted by adipose tissue [4]. This study aimed to analyze the pro-tumor potency of leptin, one of the most relevant adipokines, in relation to neuroblastoma cells and explore the pharmacological potential of cyanoenone-bearing triterpenoids in modulating this interconnection.

In addition to adipocyte-produced leptin, endogenous leptin expressed in tumors also stimulates the growth and invasiveness of malignant cells, as shown in studies of breast cancer, non-small cell lung cancer, and melanoma [37,38,39]. Based on these findings, we decided to investigate whether leptin is expressed in human neuroblastoma tissues. The performed reanalysis of transcriptomic data from GSE49710 clearly demonstrates that intratumoral expression of leptin markedly increases with neuroblastoma grade (Figure 2A) and is correlated with poor survival in neuroblastoma patients (Figure 2B). We believe that leptin, produced by tumor cells or their microenvironment, may act as an autocrine and paracrine factor, enhancing the growth and invasion of neuroblastoma. Indeed, treatment of Neuro2a mouse neuroblastoma cells with leptin increased their proliferation (Figure 2C), which is consistent with the previously reported mitogenic effect of leptin on SH-SY5Y human neuroblastoma cells [11,12]. Further exploration of the pro-metastatic potency of leptin revealed its stimulatory action on cell motility (Figure 2D) and its inhibitory effect on the adhesion of Neuro2a cells to ECM proteins (Figure 2G). Observed changes may reflect the initial stage of neuroblastoma metastasis, wherein tumor cells lose their adhesion to the basement membrane and invade the circulation [40]. In many cancers, this process is associated with the EMT of tumor cells, during which the cells acquire a more motile and invasive mesenchymal phenotype [41]. Despite originating from neuroectodermal cells, NC-derived tumor cells can also undergo a transition to a mesenchymal phenotype. However, the regulation of this process differs from that of epithelial carcinomas [42]. The evaluation of EMT status in leptin-treated Neuro2a cells surprisingly demonstrated that leptin partially reversed the phenotype of Neuro2a cells to an epithelial-like state, down-regulating mesenchymal marker vimentin and stimulating expression of the epithelial marker ZO-1 (Figure 2H). Additionally, no observed changes in cell morphology, such as an increase in cell size or neurite length, previously described for EMT in neuroblastoma [43,44], were found in Neuro2a cells after leptin treatment (Figure 2E,F). Thus, the obtained results clearly indicate that the revealed pro-metastatic effect of leptin in neuroblastoma cells is EMT-independent.

Previously, we showed that cyanoenone-bearing triterpenoid SM and its derivatives effectively inhibit the proliferation of KELLY and Neuro2a neuroblastoma cells at submicromolar concentrations [17] and exhibit an obvious suppressive effect on a range of metastasis-associated characteristics of tumor cells, such as EMT, motility, invasion, and clonogenicity [17,18,45]. Based on these facts, we decided to explore whether SM affects leptin-mediated processes in Neuro2a cells. Additionally, the role of the methyl ester group at the C-30 position of SM in this effect was additionally evaluated by testing S and SAO bearing hydroxyl and bulky pivalic amidoxime groups at this position, respectively (Figure 1). The obtained results showed that SM, S, and SAO at non-toxic concentrations completely abolished leptin-mediated proliferation in both Neuro2a (Figure 3B) and B16 (Figure 3D) cells with similar effectiveness. This suggests that the leptin-inhibiting effect of the compounds is independent of the cell line. Additionally, the substituents at the C-30 position of the explored triterpenoids have little to no impact on this effect. The last finding is in accordance with recently reported similar potencies of SM, S, and SAO in relation to murine macrophages [46]. The obtained results provide valuable information for the further structural optimization of cyanoenone-containing triterpenoids.

Analysis of the anti-metastatic potency of SM in Neuro2a cells revealed that it had no effect on cellular motility (Figure 4A). However, SM was able to restore the adhesion of leptin-stimulated neuroblastoma cells to ECM proteins to nearly the control level (Figure 4B). Given the latter finding and the similar pro-adhesive property of SM demonstrated recently in human U87 glioblastoma cells [47], our focus shifted to its impact on the expression of key regulators of cellular adhesion. Evaluation of the effect of SM on the expression of CAMs in Neuro2a cells revealed its moderate stimulatory potency in relation to *L1cam* and *Ncam1*, playing an important role in intercellular binding and communication [48], and SM-induced downregulation of *Itga1* (Figure 4C), a known receptor for laminin and collagen [49].

Considering the relatively low susceptibility of the mentioned CAMs genes to SM treatment (changes in gene expression did not exceed 60%) (Figure 4C), we hypothesized that the observed pro-adhesive potency of SM could be mediated by its effect on the expression of vimentin, which was significantly down-regulated in leptin-stimulated Neuro2a cells (Figure 2H). Vimentin provides mechanical strength and stiffness to cells and regulates their adhesion to collagen. It binds to the cytoplasmic tail of the β1 integrin and recruits talin and paxillin, resulting in the maturation of integrin clusters and the maintenance of cell-matrix contacts [28]. Indeed, treatment of leptin-stimulated cells with SM effectively restored vimentin expression to nearly the control level (Figure 4D). We assume that SM, through the upregulation of vimentin, may promote the formation of focal adhesions in Neuro2a cells [28] and, thus, prevent their detachment from collagen and collagen-containing Matrigel (Figure 4B). However, further experimental studies are required to confirm this hypothesis. Along with vimentin, SM significantly enhanced the expression of ZO-1 (Figure 4D), which regulates tight junction assembly by integrating the actin cytoskeleton with transmembrane proteins such as claudins and occludin [50]. Considering the key role of ZO-1 in cell–cell adhesion [51], the upregulation of ZO-1 under SM treatment could additionally contribute to the pro-adhesive property of triterpenoid. This finding is in line with published data. Previously, a similar effect of SM on ZO-1 expression was demonstrated in A549 human lung adenocarcinoma cells [18].

The MAPK/ERK1/2 pathway is one of the three canonical pathways triggered by leptin [30], which is associated with the leptin-mediated growth and metastasis of various epithelial-derived cancers [38,52] and NC-derived astrocytoma [53]. Our results demonstrated that leptin increased the activation of ERK1/2 in Neuro2a cells, while SM effectively abolished this effect (Figure 5A,B). Further docking simulations revealed high affinities of SM to ERK2 and its upstream regulators, SOS and MEK1 (Figure 5C–E). This demonstrates the potent multi-targeted inhibitory effect of the triterpenoid on the MAPK/ERK1/2 signaling pathway, which is consistent with previously reported data. It was found that a quinoline-bearing derivative of ursolic acid and ilexgenin A, both pentacyclic triterpenoids, effectively inhibited the kinase activity of MEK1 [54] and formed a stable complex with the active site of ERK1/2 [55], respectively. Given that the pharmacological inhibition of MEK1 by PD98059 effectively suppressed the mitogenic activity of leptin [56] and increased the adhesiveness of glioblastoma cells to ECM [57], the ability of SM to directly interact with the components of the SOS/MEK1/ERK2 signaling axis could explain its modulatory effect on the aggressiveness of leptin-induced neuroblastoma cells. Observed upregulation of vimentin under SM treatment (Figure 4D) could also be associated with ERK1/2 activation; previously, Langlois et al. demonstrated that vimentin knockout mice exhibited an elevated phosphorylation status of ERK1/2 [58]. Considering the crucial role of the SOS/MEK1/ERK2 signaling axis in leptin signaling in various cancers (not only in neuroblastoma) [30], the obtained results encourage further investigation of SM as a potent inhibitor of the tumor-promoting potency of leptin.

### Limitations of the Study

The limitations of this study are as follows. First, because the experimental protocol in this work was applied only to murine Neuro2a cells, further in-depth studies are required to validate the results obtained using a wider range of cell lines, including human neuroblastoma cells. Second, to carefully verify the stimulatory effect of leptin on neuroblastoma aggressiveness, leptin-associated neuroblastoma progression should be further evaluated in relevant animal models (e.g., high-fat diet-fed mice).

## 4. Materials and Methods

### 4.1. Chemicals and Reagents

The chemical synthesis and characterization of soloxolone methyl (SM), soloxolone (S), and soloxolone amidoxime (SAO) by chemical and nucleic magnetic resonance analysis have been previously reported [46,59]. The compounds were dissolved at a concentration of 10 mM in DMSO and were stored at −20 °C until use. Mouse recombinant leptin was purchased from ProSpec-Tany TechnoGene Ltd. (Ness-Ziona, Israel). DAPI (4′,6-diamidino-2-phenylindole) was obtained from Thermo Fisher Scientific (Dreieich, Germany). Primary rabbit antibodies, including anti-ZO-1 (ab216880), anti-vimentin (ab92547), anti-fibronectin (ab2413), and AKT/MAPK signaling pathway antibody cocktail (ab151279), as well as HRP- and Alexa Fluor^®^ 488-conjugated secondary antibodies against rabbit IgG (ab205718 and ab150077, respectively) were purchased from Abcam (Cambridge, MA, USA).

### 4.2. Cell Lines

The murine melanoma B16 and neuroblastoma Neuro2a cell lines were obtained from the Russian Culture Collection (Institute of Cytology of the Russian Academy of Sciences (RAS), St. Petersburg, Russia). The cells were cultured in Dulbecco’s modified Eagle’s medium (DMEM) (Sigma Aldrich, St. Louis, MI, USA) supplemented with 10% (*v*/*v*) heat-inactivated fetal bovine serum (FBS) (Gibco, Grand Island, NY, USA) and antibiotic–antimycotic solution contained penicillin at 10,000 IU/mL, streptomycin at 10,000 µg/mL, and amphotericin at 25 µg/mL (MP Biomedicals, Illkirch-Graffenstaden, France). Cell lines were maintained at 37 °C and 5% CO_2_ in a humidified atmosphere.

### 4.3. Biological Evaluations

#### 4.3.1. Cell Viability Analysis

Cell viability was measured in 96-well plates (2 × 10^4^ and 1.1 × 10^4^ cells per well for B16 and Neuro2a, respectively) after 48 h of incubation with leptin (200–800 ng/mL) and the studied compounds (0.1–0.4 µM) using the 3-(4,5-dimethylthiazol-2-yl)-2,5-diphenyltetrazolium bromide (MTT) assay. MTT was added to the wells at 0.5 mg/mL, and after 2 h of incubation, the formed formazan crystals were solubilized with DMSO. After that, the absorbance was measured at test and reference wavelengths of 570 nm and 620 nm, respectively, using a Multiscan RC plate reader (Thermo LabSystems, Helsinki, Finland).

#### 4.3.2. Flow Cytometry

For cell cycle analysis, Neuro2a cells after 48 h incubation with leptin (400 ng/mL) and SM (0.1 µM) were detached from 24-well plate with TrypLE Express (Gibco, Grand Island, NY, USA) and fixed in cold 70% ethanol. The cells were resuspended in PBS, and DAPI was added at 10 µg/mL.

Cell marker expression was assessed using the FIX and PERM^™^ Cell Permeabilization Kit (Invitrogen, Frederick, MD, USA). Leptin/SM-treated Neuro2a cells were collected in a 1.5 mL tube using TrypLE and fixed with Reagent A for 15 min at room temperature (RT). Cells were then washed with PBS containing 5% FBS (5% FBS/PBS) and incubated with antibodies against vimentin, fibronectin, and ZO-1 (1:100 in Reagent B) for 1 h at RT. After washing with 5% FBS/PBS, the cells were stained with Alexa Fluor^®^ 488-conjugated secondary antibody (1:2000 in 2% FBS/PBS) for 30 min in the dark at RT, washed again, and resuspended in 2% FBS/PBS.

Flow cytometry analysis was performed using the ACEA NovoCyte flow cytometer (ACEA Biosciences Inc, San Diego, CA, USA). For each sample, 10,000 events were acquired.

#### 4.3.3. Wound Healing Assay

Neuro2a cells (4 × 10^5^ cells per well) were seeded in a 24-well plate and cultured until reaching approximately 90% confluence. After that, vertical wounds were made on the cell monolayers with a sterile 10 µL pipette tip, and the detached cells were washed away with PBS before adding serum-free DMEM containing leptin (200, 400 ng/mL) and SM (0.1 µM). Sequential images of wounds were captured using a camera-equipped ZEISS Primo Vert microscope (Carl Zeiss Microscopy GmbH, Jena, Germany) at 0, 24, and 48 h time points, and wound area reduction was calculated using ImageJ 1.53k software (National Institutes of Health, Bethesda, MD, USA).

#### 4.3.4. Morphological Analysis

A total of 5 × 10^3^ Neuro2a cells were incubated with leptin (400 ng/mL) for 48 h in a 96-well plate. At least five independent microscopic fields were photographed using a ZEISS Primo Vert microscope (Carl Zeiss Microscopy GmbH, Jena, Germany). Cell area was calculated using ImageJ software.

#### 4.3.5. Neuritogenesis Assay

A total of 1 × 10^5^ Neuro2a cells were plated in a 6-well plate and treated with leptin (400 ng/mL) for 48 h. After fixation with 4% formaldehyde, the cells were stained with crystal violet dye (0.1% *w*/*v*) and photographed using a ZEISS Primo Vert microscope (Carl Zeiss Microscopy GmbH, Jena, Germany). Quantification of neurite length was performed using the ImageJ plugin NeuronJ [60].

#### 4.3.6. Adhesion Assay

A total of 5 × 10^5^ Neuro2a cells were seeded in a 6-well culture plate, allowed to adhere for 12 h, and then treated with leptin (400 ng/mL) and SM (0.1 µM). After a 24 h incubation period, cells were removed from the plate with TrypLE, and 10^5^ cells were placed in 96-well plates which were coated with Matrigel (BD Biosciences, Bedford, MA, USA) or rat tail collagen (Cell Applications Inc., San Diego, CA, USA). After 1 h, non-attached cells were washed away with PBS, and the number of adhered cells was determined using the MTT assay (see Section 4.3.1).

#### 4.3.7. RT-qPCR

After 24 h incubation with leptin (400 ng/mL) and SM (0.1 µM), Neuro2a cells were lysed with TRIzol Reagent (Ambion, Austin, TX, USA). Total RNA was isolated according to the manufacturer’s protocol and was quantified using a NanoDrop™ OneC Spectrophotometer (Thermo Fisher Scientific, Waltham, MA, USA). cDNA was synthesized from 4 µg of RNA template using the M-MuLV-RH revertase (Biolabmix, Novosibirsk, Russia) and oligo(dT)18 primer. RT-qPCR was performed using BioMaster SYBR Blue reagent kit (Biolabmix, Novosibirsk, Russia) and the primers listed in Appendix A. The expression of the studied genes was determined using the comparative cycle threshold method (2^−ΔΔCt^) and normalized to the housekeeping gene *GAPDH*.

#### 4.3.8. Western Blotting

Neuro2a cells (10^5^ cells per well) were treated with SM (0.1 µM) for 1 h prior to a 30 min induction with leptin (400 ng/mL), after which they were lysed with Laemmli buffer (Sigma-Aldrich, St. Louis, MO, USA). Proteins were separated in an SDS-polyacrylamide gel and were then transferred to a PVDF membrane, which was subsequently blocked by incubation with 2.5% nonfat dried milk for 1 h at RT. The membrane was incubated with the AKT/MAPK signaling pathway antibody cocktail, diluted 1:250 in 2.5% nonfat milk, overnight at 4 °C. Next day, the membranes were washed in PBS supplemented with 0.1% Tween-20 and incubated with secondary antibodies (HRP-conjugated goat anti-rabbit IgG, 1:3000) for 1 h at RT. Finally, the membranes were incubated with a chemiluminescent substrate (Abcam, Waltham, MA, USA) for 3 min and visualized using an iBright^™^ CL1500 Imaging System (Thermo Fisher Scientific, Waltham, MA, USA). Protein band intensity was quantified using ImageJ software.

### 4.4. Bioinformatics and Data Analysis

#### 4.4.1. Microarray Data Analysis

The dataset GSE49710 (498 primary neuroblastoma samples) was downloaded using the getGEO function in the R-package “GEOquery”. Data processing and visualization were performed using the “dplyr” and “ggplot2” packages, respectively. Survival analysis based on a log-rank test was performed using the “survival” package, and Kaplan–Meier survival curves were plotted using the “survminer” package.

#### 4.4.2. Molecular Docking

Crystal structures of SHP2, SOS, H-RAS, c-Raf, and MEK1 were imported from the Protein Data Bank [61]. Co-crystallized water and ligands were removed using Discovery Studio Visualizer v.21.1, after which polar hydrogen atoms and Gasteiger charges were added using AutoDockTools v.1.5.7. Molecules of SM and known inhibitors were prepared using the following procedure: the molecules were drawn in MarvinSketch v.22.1, optimized using the MMFF94 force field in Avogadro v.1.2.0, and allowed free rotation of their bonds using AutoDockTools. At the next stage, the ligands were docked to the interaction sites of known inhibitors using AutoDock Vina v.1.2.5 [62], and the docking structures were visualized in Discovery Studio Visualizer. For each protein, we searched the most reliable docking poses based on binding energies and the presence of hydrogen bonds (Appendix A).

#### 4.4.3. Statistical Analysis

The statistical analysis was performed using a two-tailed unpaired t-test using Microsoft^®^ Excel^®^ 2013 (v. 15.0.4569.1504). *p*-values < 0.05 were considered statistically significant.

## 5. Conclusions

The data presented here provide, for the first time, evidence that leptin is involved in the progression of neuroblastoma. Reanalysis of transcriptomic data showed that the expression of leptin was increased in advanced neuroblastoma and was inversely correlated with patient survival. Our findings clearly indicate that leptin significantly enhances the proliferation and motility of murine Neuro2a neuroblastoma cells, while reducing their adhesion to ECM proteins. This effect is accompanied by the activation of the MAPK/ERK1/2 signaling pathway. It was found that the semisynthetic triterpenoid SM effectively (1) abolished the mitogenic activity of leptin in neuroblastoma cells, inducing G1/S cell cycle arrest, and (2) enhanced their adhesiveness to various surfaces by up-regulating *L1cam*, *Ncam1*, vimentin, and ZO-1. The mechanistic studies revealed that the modulating effect of SM on leptin-stimulated Neuro2a cells may be mediated by its suppressive effect on ERK1/2 phosphorylation, possibly due to the direct interaction of SM with SOS, MEK1, and ERK2. Taken together, our results highlight the targeting of leptin signaling as a novel therapeutic strategy in NC-derived cancers and demonstrate that SM can be considered a promising lead compound for drug development for neuroblastoma treatment.

## Figures and Tables

**Figure 1 pharmaceuticals-16-01369-f001:**
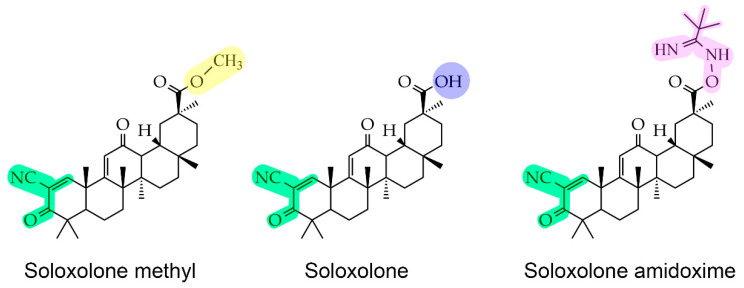
Chemical structures of soloxolone methyl (SM), soloxolone (S), and soloxolone amidoxime (SAO). The cyanoenone group, methyl ester group, hydroxyl group, and pivalic amidoxime group are marked in green, yellow, blue, and magenta, respectively.

**Figure 2 pharmaceuticals-16-01369-f002:**
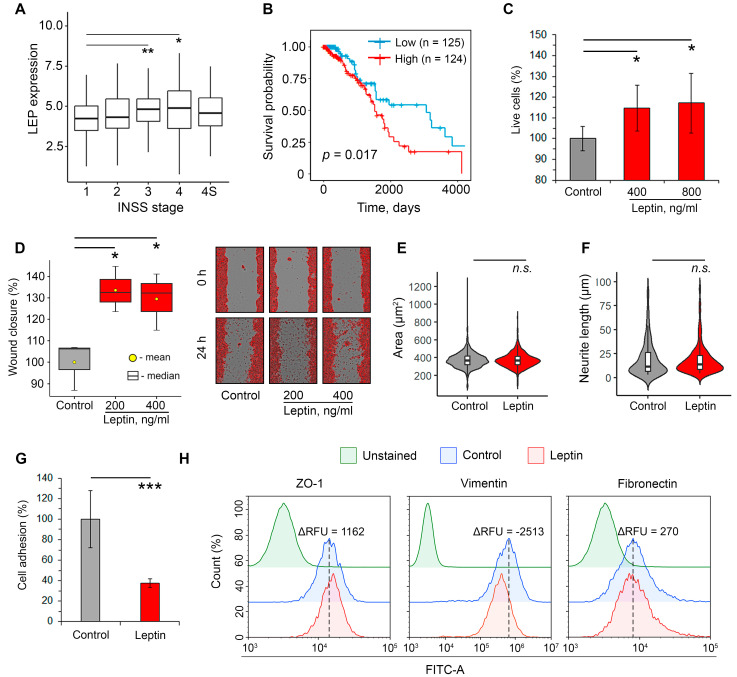
Effect of leptin on the progression of neuroblastoma. (**A**) The box plot displays the relative expression of leptin (LEP) at different INSS stages in a cohort of patients with neuroblastoma (GSE49710). (**B**) Kaplan–Meier analysis of LEP (lower quartile versus upper quartile) in patients with neuroblastoma (GSE49710). (**C**) Mitogenic effect of leptin on Neuro2a cells assessed by the MTT assay after 48 h of incubation. (**D**) Neuro2a cell motility assessed by a wound healing assay after 24 h of leptin stimulation. (**E**,**F**) Leptin had no effect (400 ng/mL) on the size (**E**) or neurite length (**F**) of Neuro2a cells, as evaluated by light microscopy. (**G**) Adhesion of Neuro2a cells to Matrigel after 24 h of incubation with leptin. (**H**) Flow cytometric determination of the expression of ZO-1, vimentin, and fibronectin in Neuro2a cells after 48 h of incubation with leptin. Data are represented as the mean  ±  SD. *** *p* < 0.001, ** *p*  <  0.01, * *p*  <  0.05, *n.s.*—non-significant vs. control.

**Figure 3 pharmaceuticals-16-01369-f003:**
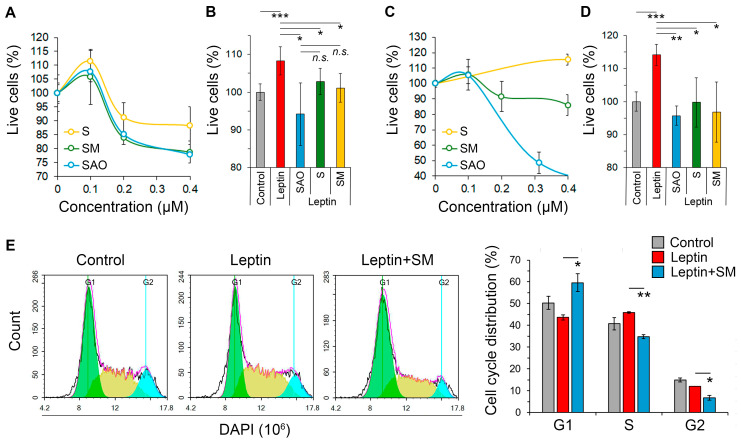
Disruption of the leptin-mediated proliferation of neuroblastoma cells by SM administration. (**A**,**C**) The cytotoxicity of SM, S, and SAO against Neuro2a cells (**A**) and B16 cells (**C**) assessed using the MTT assay after 48 h of incubation. (**B**,**D**) Triterpenoid-induced blockade of the leptin-mediated stimulation of neuroblastoma (**B**) and melanoma (**D**) cell proliferation. B16 and Neuro2a cells were incubated with leptin (400 ng/mL) and triterpenes (0.1 µM) for 48 h, and their viability was assessed using the MTT assay. (**E**) Cell cycle phase analysis of Neuro2a cells exposed to leptin and SM for 48 h. Data are represented as the mean  ±  SD. *** *p*  <  0.001 vs. control, ** *p*  <  0.01, * *p*  <  0.05 vs. leptin group; *n.s.*—non-significant vs. SAO group.

**Figure 4 pharmaceuticals-16-01369-f004:**
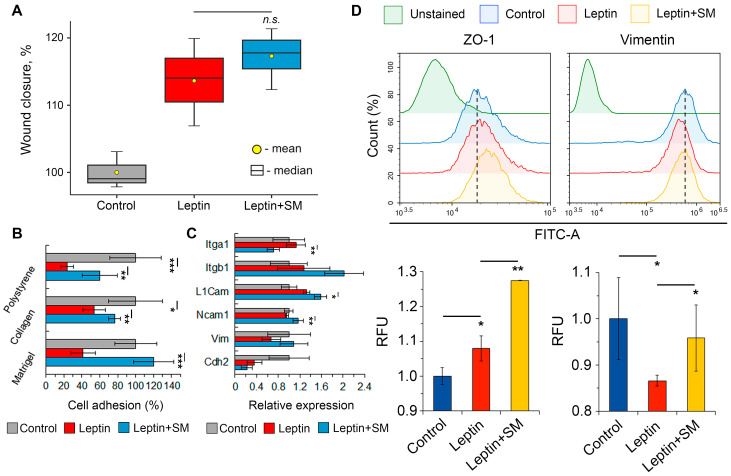
Antimetastatic activity of SM in leptin-mediated neuroblastoma. (**A**) Wound closure on Neuro2a monolayers after 48 h-induction with leptin and SM. (**B**) Adhesion of Neuro2a cells to Matrigel, collagen, and culture plastic (polystyrene) after 24 h of incubation with leptin and SM. (**C**) mRNA expression of CAMs in Neuro2a cells assessed by RT-qPCR after 48 h of leptin and SM induction. (**D**) Expression of ZO-1 and vimentin in Neuro2a cells after 48 h induction with leptin and SM assessed by flow cytometry. Data are represented as the mean  ±  SD. *** *p* < 0.001 vs. control or leptin group, ** *p*  <  0.01 vs. control, * *p*  <  0.05 vs. control or leptin group, *n.s*.—non-significant vs. leptin group.

**Figure 5 pharmaceuticals-16-01369-f005:**
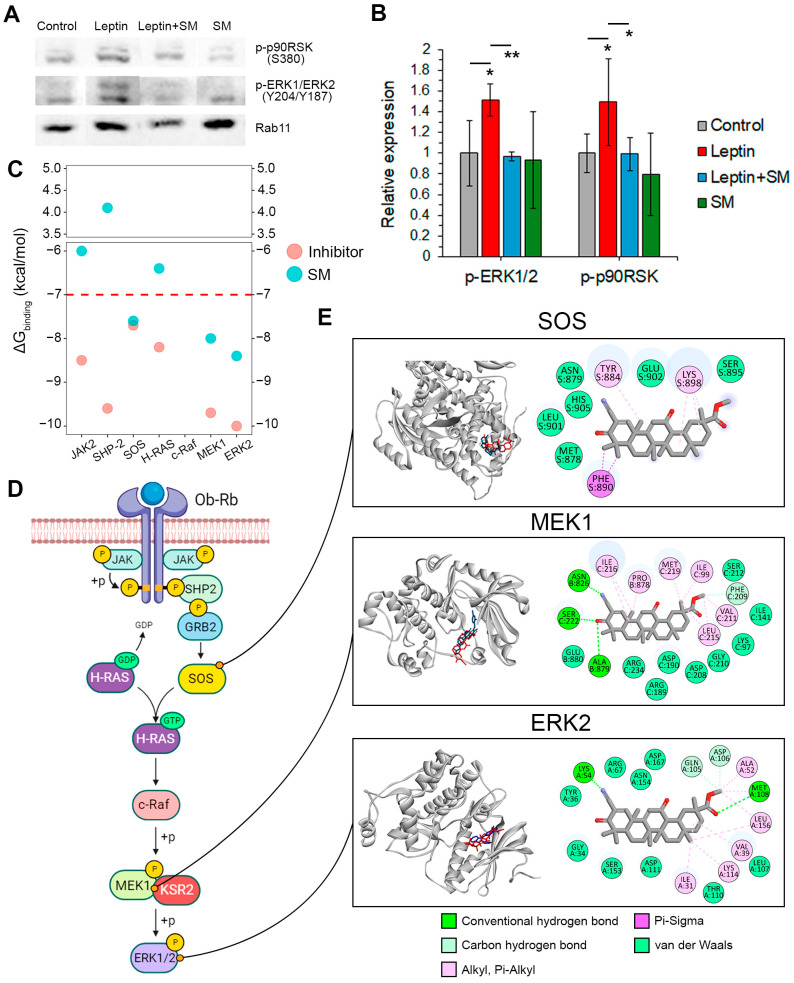
Inhibition of the MAPK/ERK1/2 signaling axis by SM. (**A**) Western blot analysis of ERK1/2 (Tyr204/Tyr187) and p90RSK (Ser380) phosphorylation in Neuro2a cells induced by SM for 1 h and then treated with leptin for 30 min. (**B**) Densiometer values for p-ERK1/2 and p-p90RSK. (**C**) Gibbs free energy (ΔG) of SM binding to SOS, MEK1, and ERK2 calculated using AutoDock Vina. The binding energies of known inhibitors are given for comparison. The red line indicates the affinity threshold, below which binding is considered strong. (**D**) Schematic representation of leptin-induced activation of MAPK/ERK signaling. (**E**) Predicted interactions of SM with SOS, MEK1, and ERK2 visualized using Discovery Studio Visualizer. Red and blue molecules represent SM and known inhibitors, respectively. Data are represented as the mean  ±  SD. ** *p*  <  0.01 vs. leptin group, * *p*  <  0.05 vs. control or leptin group.

## Data Availability

On reasonable request, the corresponding author will provide the data generated and/or analyzed during this study.

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
