# Peer review of "Soloxolone Methyl Reduces the Stimulatory Effect of Leptin on the Aggressive Phenotype of Murine Neuro2a Neuroblastoma Cells via the MAPK/ERK1/2 Pathway"

_pharmaceuticals, 2023, doi:10.3390/ph16101369_

Round 1
Reviewer 1 Report
The study is interesting and shows that soloxolone methyl (SM) can inhibit the MAPK/ERK1/2 signaling pathway and inhibit the pro-tumorigenic roles of leptin in cancer cell lines.
Why did the authors decide to pursue studies only with SM after fig. 3C? SAO also showed lower cell viability in both neuroblastoma and melanoma cell lines (added together with leptin) as shown in Fig 3B and C. Why was SAO and S not included in the study further? I also have a concern regarding the western blot image in fig. 5A. It can be clearly seen that the blot image has been cropped and put together as panel A in the figure. The authors have provided the full blot images used to create fig 5A but they should clearly indicate the exact positions of the molecular weight markers in the full blots. I recommend a major revision for this manuscript.
Author Response
Dear Reviewer #1,
We appreciate your valuable comments and suggestions that helped us improve the manuscript. Let us respond to your remarks.
Why did the authors decide to pursue studies only with SM after fig. 3C? SAO also showed lower cell viability in both neuroblastoma and melanoma cell lines (added together with leptin) as shown in Fig 3B and C. Why was SAO and S not included in the study further?
Authors: Our study focused mainly on SM, which can be considered as a promising drug candidate according to a number of studies [1–3]. The purpose of comparing SM with its derivatives, S and SAO, was not to identify a hit compound, but to find out whether the methyl group at C-30 affects the activity of SM. The differences you mentioned between SM and SAO lacked statistical significance according to a two-tailed unpaired t-test, so we concluded that substituents at C-30 position have little to no effect on the activity of triterpenoids with respect to the mentioned neuroblastoma cell characteristics. Considering the similar bioactivities of SM, S, and SAO (Fig. 3B, D) and the previously reported ability of SM to block the aggressiveness of murine B16 melanoma cells (neural crest-derived cells) [1], we decided to select SM for further studies. To clarify this, a proposal for SM selection has been added to the manuscript (see p. 6, lines 10-12).
I also have a concern regarding the western blot image in fig. 5A. It can be clearly seen that the blot image has been cropped and put together as panel A in the figure. The authors have provided the full blot images used to create fig 5A but they should clearly indicate the exact positions of the molecular weight markers in the full blots.
Authors: Dear Reviewer #1, thank you very much for this valuable comment. We have corrected the full blot image and presented it below (Fig. R1) and additionally in the supplementary materials (Please see, Fig. S2). We used photos with longer exposure time to analyze the intensity of p-ERK1/ERK2 (Fig. R1A) and shorter exposure time to analyze p-p90RSK and Rab11 (Fig. R1B). Unfortunately, of all the molecular weight markers (PageRuller™ Ladder cat. 26619, Thermo Fisher, USA), only two red bands 70 and 25 kDa are visible in the final blot image (Fig. R1A). However, as shown in Fig. R1A, all bands are clearly visible, and the comparison of the obtained blot images with the example image in the manufacturer’s brochure (AKT/MAPK Signaling Pathway Antibody Cocktail ab151279, Cambridge, UK) (Fig. R1C) allows us to clearly determine their location:
- The p-p90RSK band roughly corresponds approximately to the 70 kDa marker band (Fig. R1A), which is close to the relative positions we can observe for p-p90RSK and the 75 kDa marker band in the example image (Fig. R1C).
- The p-RPS6 band is higher than that of the 25 kDa marker band and the Rab11 band is close to the 25 kDa marker band, corresponding to the relative band positions of p-RPS6 and Rab11 in the example image (Fig. R1C).
- The two p-ERK1/ERK2 bands are located above p-RPS6 in the same way as in the example image (Fig. R1A, R1C).
Thus, based on the analysis performed, we can confirm the correctness of the identification of the analyzed proteins.
Fig. R1. Western blot analysis of p-p90RSK and p-ERK1/ERK2 expression in Neuro2a cells induced by SM and leptin. (A) Representative image used to analyze p-ERK1/ERK2 intensity. Black arrows indicate the positions of molecular weight markers (PageRuller™ Ladder cat. 26619, Thermo Fisher, USA). (B) Representative image used to analyze p-p90RSK and Rab11 intensity. (C) Example image of blots from the manufacturer's brochure (ab151279, Cambridge, UK).
We hope that corrected version of the manuscript will be acceptable for publication in the Pharmaceuticals.
Sincerely,
On behalf of all authors,
Dr. Andrey Markov

Reviewer 2 Report
The manuscript tested: 1) whether leptin is capable of increasing the aggressiveness of neuroblastoma, 2) whether cyanoenone-bearing triterpenoids can suppress the tumor-promoting effect of leptin on neuroblastoma cells. They found that SM Reduced the stimulatory Effect of Leptin on the aggressive phenotype of neuroblastoma cells via the MAPK/ERK1/2 Pathway. These provide useful information to the field of neuroblastoma. Major concerns include:
1. In Fig 2A, does the expression of leptin refer to its mRNA or protein? It is confused. If it is protein, please add the immunostaining of Leptin. If it is mRNA, the relation between the leptin mRNA of neuroblastoma and obesity is difficultly understood. please explain.
2. Please the photos of cultured cells related to Fig2D-G.
3. It is better to shorten the manuscript.
It is better to shorten the manuscript.
Author Response
Dear Reviewer #2,
Thank you for taking the time to read and thoroughly analyze our article. We revised the manuscript according to your comments, and, please, let us respond to them.
- In Fig 2A, does the expression of leptin refer to its mRNA or protein? It is confused. If it is protein, please add the immunostaining of Leptin. If it is mRNA, the relation between the leptin mRNA of neuroblastoma and obesity is difficultly understood. please explain.
Authors: Dear Reviewer #2, thank you for pointing out the uncertainty we have left in the text. Leptin expression in Fig. 2A refers to mRNA because we reanalyzed the cDNA microarray data (GSE49710). To make this clearer, we have included this information in the Results section (see p. 3, lines 14 and 20).
In addition to leptin produced by adipocytes, leptin has been shown to be expressed in a number of tumors, including breast cancer, non-small cell lung cancer, and melanoma [1–3]. To avoid confusion for the reader, we have added a sentence describing tumor endogenous leptin in the Discussion section (see p. 9, lines 21-24). Our re-analysis of the microarray data showed an increase in leptin expression in tumor tissue during neuroblastoma progression (Fig. 2A), which is also associated with reduced patient survival (Fig. 2B). Based on these facts, we speculate that leptin produced by tumor cells or their microenvironment may act as an autocrine and paracrine factor to enhance the aggressiveness of neuroblastoma. However, we do not exclude the influence of obesity or systemic leptin on neuroblastoma because they have prognostic significance in some cancers such as breast cancer, pancreatic adenocarcinoma, and melanoma [4,5]. However, we cannot confirm this because the microarray data (GSE49710) do not include information on patient weight or serum leptin levels. Therefore, these relationships should be investigated in future studies.
- Please the photos of cultured cells related to Fig2D-G.
Authors: Corrected. We have added photos of wounds to Fig. 2D and photos of cultured cells for Figs. 2E and 2F in the Supplementary Material (see Figs. 2D, S1A, S1B).
- It is better to shorten the manuscript.
Authors: Corrected. Dear Reviewer #2, we have tried to shorten our manuscript as much as possible so that the main results and assumptions are not lost. Part of the text in the Introduction and Results sections has been deleted (please, see p. 2, lines 12-13, lines 22-23; p. 3, line 30).
Minor editing of English language required
Authors: Corrected. With the help of our foreign colleagues, the text of the manuscript has been proofread.
We hope that this version of the manuscript will be acceptable for publication.
Thank you very much!
Sincerely,
On behalf of all authors,
Dr. Andrey Markov
References
- Gelsomino, L.; Giordano, C.; La Camera, G.; Sisci, D.; Marsico, S.; Campana, A.; Tarallo, R.; Rinaldi, A.; Fuqua, S.; Leggio, A.; et al. Leptin Signaling Contributes to Aromatase Inhibitor Resistant Breast Cancer Cell Growth and Activation of Macrophages. Biomolecules 2020, 10, 543, doi:10.3390/biom10040543.
- Li, F.; Zhao, S.; Guo, T.; Li, J.; Gu, C. The Nutritional Cytokine Leptin Promotes NSCLC by Activating the PI3K/AKT and MAPK/ERK Pathways in NSCLC Cells in a Paracrine Manner. Biomed Res. Int. 2019, 2019, 2585743, doi:10.1155/2019/2585743.
- Ellerhorst A., J.; Diwan H., A.; Dang M., S.; Uffort G., D.; Johnson K., M.; Cooke P., C.; Grimm A., E. Promotion of melanoma growth by the metabolic hormone leptin . Oncol Rep 2010, 23, 901–907, doi:10.3892/or_00000713.
- Lin, T.-C.; Hsiao, M. Leptin and Cancer: Updated Functional Roles in Carcinogenesis, Therapeutic Niches, and Developments. Int. J. Mol. Sci. 2021, 22.
- Avgerinos, K.I.; Spyrou, N.; Mantzoros, C.S.; Dalamaga, M. Obesity and cancer risk: Emerging biological mechanisms and perspectives. Metabolism 2019, 92, 121–135, doi:https://doi.org/10.1016/j.metabol.2018.11.001.

Reviewer 3 Report
Excellent work, worthy of publication. Do the authors have any proteomic data on leptin levels in neuroblastoma tumor samples? Are the leptin concentrations they used in their experimental model compatible with those observed in vivo? This aspect should be discussed.
Minor point: page 5, line19: it’s better to write ”… were evaluated in Neuro2a cells.”
Author Response
Dear Reviewer #3,
We appreciate your consideration and interest in our work. Thank you for recommending the publication of our manuscript. We have edited the manuscript taking into account your comments and provide answers to your questions.
Do the authors have any proteomic data on leptin levels in neuroblastoma tumor samples?
Authors: To the best of our knowledge, there are no available data on leptin protein expression in neuroblastoma tumors. However, leptin is expressed and secreted by mouse Neuro2a and human SH-SY5Y neuroblastoma cells as demonstrated by Western blot and ELISA, respectively [1,2]. Thus, this issue requires further study.
Are the leptin concentrations they used in their experimental model compatible with those observed in vivo? This aspect should be discussed.
Authors: Corrected. In our experiments, we chose leptin concentrations of 200 ng/ml and 400 ng/ml because these levels of leptin have been published to stimulate the proliferative and migratory potential of tumor cells [3–6]. Moreover, these leptin concentrations correspond to those found in the serum of high-fat diet (HFD)-induced obese mice, which we could choose as an in vivo model to study the association between obesity and neuroblastoma. For example, there are studies reporting serum leptin concentrations of 100 ng/ml, 120 ng/ml, and 370 ng/ml in HFD-induced mice [7–9]. For greater clarity, we have added this information to our manuscript (please, see p. 3, lines 26-28).
Minor point: page 5, line19: it’s better to write ”… were evaluated in Neuro2a cells.”
Authors: Corrected. Please, see p. 5, lines 18–19.
We hope that this version of the manuscript will be acceptable for publication.
Thank you very much!
Sincerely,
On behalf of all authors,
Dr. Andrey Markov
References
- Marwarha, G.; Claycombe, K.; Schommer, J.; Collins, D.; Ghribi, O. Palmitate-induced Endoplasmic Reticulum stress and subsequent C/EBPα Homologous Protein activation attenuates leptin and Insulin-like growth factor 1 expression in the brain. Cell. Signal. 2016, 28, 1789–1805, doi:10.1016/j.cellsig.2016.08.012.
- Marwarha, G.; Dasari, B.; Ghribi, O. Endoplasmic reticulum stress-induced CHOP activation mediates the down-regulation of leptin in human neuroblastoma SH-SY5Y cells treated with the oxysterol 27-hydroxycholesterol. Cell. Signal. 2012, 24, 484–492, doi:https://doi.org/10.1016/j.cellsig.2011.09.029.
- Mendonsa, A.M.; Chalfant, M.C.; Gorden, L.D.; VanSaun, M.N. Modulation of the Leptin Receptor Mediates Tumor Growth and Migration of Pancreatic Cancer Cells. PLoS One 2015, 10, e0126686.
- Alfredo, V.D.; Miriam Daniela, Z.E.; Jose, D.B.; Eduardo, C.S.; Mercedes, C.G.; Angel, M.C.M.; Carlos, O.P.; Napoleon, N.T. Leptin induces partial epithelial-mesenchymal transition in a FAK-ERK dependent pathway in MCF10A mammary non-tumorigenic cells. Int. J. Clin. Exp. Pathol. 2017, 10, 10334–10342.
- García-Miranda, A.; Solano-Alcalá, K.A.; Montes-Alvarado, J.B.; Rosas-Cruz, A.; Reyes-Leyva, J.; Navarro-Tito, N.; Maycotte, P.; Castañeda-Saucedo, E. Autophagy Mediates Leptin-Induced Migration and ERK Activation in Breast Cancer Cells. Front. Cell Dev. Biol. 2021, 9, 644851.
- Duan, L.; Lu, Y.; Xie, W.; Nong, L.; Jia, Y.; Tan, A.; Liu, Y. Leptin promotes bone metastasis of breast cancer by activating the SDF-1/CXCR4 axis. Aging (Albany. NY). 2020, 12, 16172–16182, doi:10.18632/aging.103599.
- Trottier, M.D.; Naaz, A.; Li, Y.; Fraker, P.J. Enhancement of hematopoiesis and lymphopoiesis in diet-induced obese mice. Proc. Natl. Acad. Sci. 2012, 109, 7622–7629, doi:10.1073/pnas.1205129109.
- Lee, H.S.; Jeon, Y.E.; Jung, J.I.; Kim, S.M.; Hong, S.H.; Lee, J.; Hwang, J.S.; Hwang, M.O.; Kwon, K.; Kim, E.J. Anti-obesity effect of Cydonia oblonga Miller extract in high-fat diet-induced obese C57BL/6 mice. J. Funct. Foods 2022, 89, 104945, doi:https://doi.org/10.1016/j.jff.2022.104945.
- de Melo, T.S.; Lima, P.R.; Carvalho, K.M.M.B.; Fontenele, T.M.; Solon, F.R.N.; Tomé, A.R.; de Lemos, T.L.G.; da Cruz Fonseca, S.G.; Santos, F.A.; Rao, V.S.; et al. Ferulic acid lowers body weight and visceral fat accumulation via modulation of enzymatic, hormonal and inflammatory changes in a mouse model of high-fat diet-induced obesity. Brazilian J. Med. Biol. Res. 2017, 50, e5630, doi:10.1590/1414-431X20165630.

Reviewer 4 Report
This study shows that leptin has stimulatory effects for aggressive tumors growth and soloxolone methyl and its derivatives could suppress them. The story is theoretical and the methods used are appropriate. However I have some concerns.
1. SM derivatives might not have toxic effects significantly on Neuro2a cells at the concentration of 0.1uM as seen Fig.3A. But the mean numbers (%) of live cells in treatment of 0.1uM SM derivatives look around 90%, the decrease ratio of which seems like as same tendency as those seen in Fig.3B. I think that the lower concentration, at which the live cells would be almost 100%, should be tested or that the effects of SM derivatives alone (without leptin) should be tested in each experiment.
2. The max values of the vertical axes in Fig.3D flowcytometry chart left and middle are about 250, while that in Fig3D right is about 80. Was the result in Leptin+SM evaluated fairly?
3. “Cell cycle dustribution” of Fig.3 right is “Cell cycle distribution”.
4. The control cells’ data of Fig4C are necessary.
Author Response
Dear Reviewer #4,
We sincerely appreciate the insights and guidance you have provided about our work. After making several changes to the manuscript based on your advice, we provide responses to your comments.
- SM derivatives might not have toxic effects significantly on Neuro2a cells at the concentration of 0.1uM as seen Fig.3A. But the mean numbers (%) of live cells in treatment of 0.1uM SM derivatives look around 90%, the decrease ratio of which seems like as same tendency as those seen in Fig.3B. I think that the lower concentration, at which the live cells would be almost 100%, should be tested or that the effects of SM derivatives alone (without leptin) should be tested in each experiment.
Authors: Dear Reviewer #4, thank you for this valuable comment. Indeed, the large standard deviations shown in Figure 3A may give a false impression of the toxicity of compounds used at a concentration of 0.1 μM. To reduce high dispersion in the cytotoxic data, we performed additional MTT experiments for Neuro2a and B16 cells to obtain more reliable results. As shown in the revised Fig. 3, SM, S, and SAO at 0.1 μM were nontoxic and, furthermore, slightly increased the viability of both cell lines (Please, see Fig 3A, 3C). These results are consistent with previous studies of CDDO-Me, a structural analog of SM, which also showed a slight proliferation-stimulating effect at nanomolar concentrations [1,2] probably related to the activation of Nrf2 cytoprotective pathway [3–5]. Thus, the results obtained clearly indicated that the observed blockade of the mitogenic effect of leptin by SM, S, and SAO was not due to the toxicity of the compounds (Please, see Fig. 3B, 3D).
- The max values of the vertical axes in Fig.3D flowcytometry chart left and middle are about 250, while that in Fig3D right is about 80. Was the result in Leptin+SM evaluated fairly?
Authors: Corrected. Dear Reviewer #4, we appreciate you pointing out this omission. We have found and corrected a gating error for the Leptin+SM sample and brought it in line with samples from other groups (see Fig. 3E). This resulted in minor changes in cell cycle distribution and statistical significance in the Leptin+SM group, but did not affect the overall conclusion that SM blocks the G1/S transition.
- “Cell cycle dustribution” of Fig.3 right is “Cell cycle distribution”.
Authors: Corrected. Thanks a lot! Please, see Fig. 3E.
- The control cells’ data of Fig4C are necessary.
Authors: Corrected. We added control group to Fig. 4C and added their description to the Results section (Please, see p. 6, lines 35-41, Fig. 4C).
We hope that this version of the manuscript will be acceptable for publication.
Thank you very much!
Sincerely,
On behalf of all authors,
Dr. Andrey Markov
References
- Qin, Y.; Deng, W.; Ekmekcioglu, S.; Grimm, E.A. Identification of unique sensitizing targets for anti-inflammatory CDDO-Me in metastatic melanoma by a large-scale synthetic lethal RNAi screening. Pigment Cell Melanoma Res. 2013, 26, 97–112, doi:https://doi.org/10.1111/pcmr.12031.
- Wang, X.Y.; Zhang, X.H.; Peng, L.; Liu, Z.; Yang, Y.X.; He, Z.X.; Dang, H.W.; Zhou, S.F. Bardoxolone methyl (CDDO-Me or RTA402) induces cell cycle arrest, apoptosis and autophagy via PI3K/AKT/mTOR and p38 MAPK/ErK1/2 signaling pathways in K562 cells. Am. J. Transl. Res. 2017, 9, 4652–4672.
- Borella, R.; Forti, L.; Gibellini, L.; De Gaetano, A.; De Biasi, S.; Nasi, M.; Cossarizza, A.; Pinti, M. Synthesis and anticancer activity of CDDO and CDDO-me, two derivatives of natural triterpenoids. Molecules 2019, 24, 4097, doi:10.3390/molecules24224097.
- Markov, A. V; Sen’kova, A. V; Babich, V.O.; Odarenko, K. V; Talyshev, V.A.; Salomatina, O. V; Salakhutdinov, N.F.; Zenkova, M.A.; Logashenko, E.B. Dual Effect of Soloxolone Methyl on LPS-Induced Inflammation In Vitro and In Vivo. Int. J. Mol. Sci. 2020, 21.
5. Markov, A. V.; Kel, A.E.; Salomatina, O. V.; Salakhutdinov, N.F.; Zenkova, M.A.; Logashenko, E.B. Deep insights into the response of human cervical carcinoma cells to a new cyano enone-bearing triterpenoid soloxolone methyl: A transcriptome analysis. Oncotarget 2019, 10, 5267–5297, doi:10.18632/oncotarget.27085.

Reviewer 5 Report
In the context of this manuscript, the authors endeavor to assess the potential of soloxonyl methyl to mitigate the aggressive phenotype observed in neuroblastoma cells. This objective is pursued through the attenuation of leptin's stimulatory impact on neuroblastoma cells via modulation of the MAPK/ERK pathway. While the subject matter of the article holds inherent intrigue, certain shortcomings with regard to the stipulated quality criteria are discernible.
A noteworthy concern pertains to the exclusive reliance on Neuro2a cells, an established mouse neuroblastoma cell line, as the focal experimental model. This restriction to murine cellular models warrants scrutiny, as it potentially limits the translational applicability of the findings to human neuroblastoma contexts. Neuro2a cells, while informative, present intrinsic dissimilarities in their cellular behavior and signaling cascades when compared to their human-derived counterparts. Consequently, extrapolating the observed outcomes from murine models to the human neuroblastoma scenario requires circumspection, given the intricate divergence between species-specific cellular responses.
To enhance the robustness and generalizability of the study's conclusions, it is recommended that future investigations be conducted with a more diverse array of human-derived neuroblastoma cell lines. Incorporating at least three distinct human-derived neuroblastoma cell lines, characterized by divergent genetic backgrounds and phenotypic behaviors, would contribute substantially to the comprehensive evaluation of soloxonyl methyl's efficacy in the context of human neuroblastoma. By engaging a range of human cell lines, the study would foster a more nuanced understanding of the therapeutic potential and limitations of soloxonyl methyl in countering the aggressive attributes of human neuroblastoma cells.
In summation, while the current article stands as a thought-provoking endeavor, its foundational reliance on the Neuro2a cell line raises queries regarding the extensibility of the findings to human neuroblastoma scenarios. The incorporation of human-derived cell lines within the experimental design is advocated as a means to bolster the study's clinical relevance and bolster the robustness of its conclusions.
Author Response
Dear Reviewer #5,
We are very grateful for your detailed feedback on our article. It allowed us to rethink the concept and theoretical significance of our work. Let us provide responses to your comments.
A noteworthy concern pertains to the exclusive reliance on Neuro2a cells, an established mouse neuroblastoma cell line, as the focal experimental model. This restriction to murine cellular models warrants scrutiny, as it potentially limits the translational applicability of the findings to human neuroblastoma contexts. Neuro2a cells, while informative, present intrinsic dissimilarities in their cellular behavior and signaling cascades when compared to their human-derived counterparts. Consequently, extrapolating the observed outcomes from murine models to the human neuroblastoma scenario requires circumspection, given the intricate divergence between species-specific cellular responses.
Authors: Dear Reviewer #5, we certainly agree with the limited transferability of our findings from animals to humans that you point out. However, we believe that our preliminary study in mouse cells has provided some interesting insights that may stimulate further work in this direction, as the effects of adipokines on neuroblastoma are largely unexplored. To avoid a broad interpretation of our results, we emphasized in the title and throughout the text that we used mouse cells only (see p. 1, lines 3, 17, 29; p. 2, line 53; p. 3, line 28; p. 13, line 40) and the need for further verification of our data on human neuroblastoma cells (p. 1, lines 30-31; p. 2, lines 53-54; p. 3, lines 1-2). Additionally, we also discussed the limitations of the study in the Discussion section (see p. 11, lines 9-15) and briefly mentioned them in the Introduction (see p. 2, lines 53-54).
To enhance the robustness and generalizability of the study's conclusions, it is recommended that future investigations be conducted with a more diverse array of human-derived neuroblastoma cell lines. Incorporating at least three distinct human-derived neuroblastoma cell lines, characterized by divergent genetic backgrounds and phenotypic behaviors, would contribute substantially to the comprehensive evaluation of soloxonyl methyl's efficacy in the context of human neuroblastoma. By engaging a range of human cell lines, the study would foster a more nuanced understanding of the therapeutic potential and limitations of soloxonyl methyl in countering the aggressive attributes of human neuroblastoma cells.
Authors: Dear Reviewer #5, thank you for providing us with methodological advice. We will certainly follow them when designing experiments in our further research on this and related topics to increase the applicability of our results.
We hope that this version of the manuscript with the corrected topic will be acceptable for publication.
Thank you very much!
Sincerely,
On behalf of all authors,
Dr. Andrey Markov

Round 2
Reviewer 1 Report
The authors have satisfactorily addressed my concerns, I recommend accepting the revised manuscript for publication. Thank you
Author Response
Dear Reviewer #1,
Thank you very much for taking the time to review our work and for your important comments, which have greatly improved our manuscript. We hope that our findings will be useful to researchers working in the field of neuroblastoma and the relationship between metabolic dysregulation and tumor transformation.
Sincerely,
On behalf of all authors,
Dr. Andrey Markov

Reviewer 5 Report
The author has duly incorporated justifiable modifications in response to my previous feedback. I presently have no additional comments to provide on this manuscript.
Author Response
Dear Reviewer #5,
We are very grateful for your valuable comments and remarks, which made our manuscript more understandable. We really hope that the results of our study will be useful for researchers studying the pathogenesis of neuroblastoma and the relationship between obesity and tumor transformation.
Sincerely,
On behalf of all authors,
Dr. Andrey Markov
